# Children with autism spectrum disorder and attention deficit hyperactivity disorder have evidence of sensory nerve degeneration

Hoda Gad[1], Abdulla AlObaidi[1], Madeeha Kamal[2], Saba F. Elhag[2], Marian Aden[3], Adnan Khan[1], Areej Baraka[1], Srividya Bennabhaktula[1], Sara Joseph[1], Aisha Yaseen Alhamadi[4], Mohammed A. Tolefat[4], Abdulla A. Alhothi[3], Rayaz A. Malik[1,5]*

1 Research Department, Weill Cornell Medicine-Qatar, Doha, Qatar, 2 Metabolic and Mendelian Disorders Department, Sidra Medicine, Doha, Qatar, 3 Pediatric department, Hamad Medical Corporation, Doha, Qatar, 4 Shafallah Center for Persons with Disabilities, Doha, Qatar, 5 Institute of Cardiovascular Medicine, University of Manchester, Manchester, United Kingdom

* ram2045@qatar-med.cornell.edu

## Abstract

An increasing body of evidence supports the role of altered sensory processing in autism spectrum disorder (ASD) and attention-deficit/hyperactivity disorder (ADHD). This exploratory study undertook corneal confocal microscopy (CCM) to assess for differences in corneal nerve fiber density (CNFD), branch density (CNBD) and fiber length (CNFL) in relation to ASD severity in children with ASD+ADHD (n=21), ASD (n=6) and healthy controls (HC) (n=15). CNBD was significantly lower in children with ASD+ADHD (P=0.002) and ASD (P<0.001) and CNFL was significantly lower in children with ASD (P=0.02) compared to HC. However, CNFD (mean difference=2.28, 95% CI [−5.84, 10.40], P>0.99) CNBD (mean difference=14.1, 95% CI [−8.22, 36.4], p=0.37) and CNFL (mean difference=2.46, 95% CI [−2.28, 7.20], p=0.61) did not differ between ASD+ADHD and ASD. CNFD (P=0.876), CNBD (P=0.405) and CNFL (P=0.606) did not correlate with the severity of ASD. Corneal confocal microscopy reveals sensory nerve degeneration in children with ASD+ADHD and ASD alone compared to controls. Larger studies integrating CCM with sensory and cognitive assessment are required to determine the utility of CCM as a clinical screening strategy for neurodegneration in ASD, ADHD and ASD-ADHD combined.

## Introduction

Autism Spectrum Disorder (ASD) is a neurodevelopmental condition defined by altered social communication and interaction with restricted and repetitive behaviour or interests. It can vary in its presentation, may be associated with anxiety and depression and shaped by age, gender, race, ethnicity, and developmental stage [1]. Attention deficit hyperactivity disorder (ADHD) is also a neurodevelopmental disorder characterized

which permits unrestricted use, distribution, and reproduction in any medium, provided the original author and source are credited.

**Data availability statement:** All relevant data are within the paper and its Supporting Information files.

**Funding:** This study was funded by the Autism Research Institute.

**Competing interests:** None.

by persistent inattention, hyperactivity, and impulsivity that disrupts daily functioning [2]. Both ASD and ADHD have a high chance of co-occurrence with deficits in social communication and interaction, restricted, repetitive behaviour [3], inattention, hyperactivity, and impulsivity, respectively [4]. Most research has focused on brain-centric mechanisms in ASD and ADHD. However, there is increasing recognition of abnormal sensory responses in ASD [5–10] which relate to quantitative autism traits [7] such that they constitute one of the four sub-criteria under the restricted, repetitive behaviors domain in the Diagnostic and Statistical Manual of Mental Disorders, Fifth Edition (DSM-5) [11]. Sensory alterations have been documented using clinical instruments such as the Autism Diagnostic Interview-Revised (ADI-R) scale and sensory abnormalities may underlie agitation in ADHD or interpreted as atypical sensory behavior in ASD [12].

Participants with ASD have normal pain detection thresholds and pain tolerance, but report diminished subjective pain intensity and heightened sensitivity consistent with variability in nociceptive processing and cognitive pain modulation [13]. Hyperresponsiveness of the auditory domain has been shown in 43.8% of children with ASD with hyperresponsiveness, hyporesponsiveness, and sensory-seeking behavior in the visual and proprioceptive domains [8]. Intraepidermal nerve fibre density was reduced in 53.1% and thermal thresholds were elevated in 28% of adults with ASD [14].

Ophthalmologic abnormalities occur in 48.4% of patients with ASD and include hyperopia, astigmatism and myopia with strabismus in 15.4% [15]. Youth with ASD have an increased prevalence of amblyopia, anisometropia, astigmatism and hypermetropia [16]. Corneal confocal microscopy (CCM) is a rapid non-invasive ophthalmic imaging technique that has identified corneal nerve loss in diabetic [17] and other peripheral neuropathies [18], dementia [19], multiple sclerosis [20] epilepsy [20], Parkinson's disease [21] and ASD [22].

A growing body of evidence shows altered sensory processing in ADHD [4,23–27]. In adults with ADHD, self-reported tactile sensitivity was associated with symptom severity, a lower tolerable threshold for electrical radial nerve stimulus, and greater reduction in cortical SEP amplitudes [4]. In a resting-state functional magnetic resonance imaging (rs-fMRI) study, adults with ADHD had an increased pattern of functional connectivity between sensory seed regions and the rest of the brain and decreased connectivity between sensory seeds and the default-mode network which was related to clinical severity of ADHD [25]. Individuals with higher levels of ADHD traits report hyper- and hypo-sensitivity, determined with the Glasgow Sensory Questionnaire (GSQ) and Adult ADHDH Self Report Scale (ASRS) [28].

Recent studies have shown peripheral [14] and corneal [22] nerve loss and correlations between sensory and social symptoms in individuals with ASD [29]. The aim of this pilot study was to assess for differences in sensory nerve degeneration between children with ADHD and ASD compared to ASD alone.

## Methods

### Participants and study design

This was a pilot study involving participants with ASD+ADHD (n=21), ASD-alone (n=6) and 15 healthy controls aged 7–18 years from Hamad Medical Corporation

(HMC) and Shafallah Center for Persons with Disabilities, recruited between June 2021 to October 2024. Parents of eligible participants were informed of the study by pediatricians specializing in ASD and ADHD. Healthy controls (HC) were recruited from children attending the pediatric clinic for follow up of acute conditions, without neurodevelopmental or chronic medical conditions.

Participants had mild to moderate ASD, as CCM was not possible in non-verbal individuals with severe ASD. Participants with any other cause of neuropathy, malignancy, deficiency of $B_{12}$ or folate, chronic renal failure, liver failure, connective tissue, or systemic disease (rheumatoid arthritis, systemic lupus erythematosus, dermatomyositis, systemic scleroderma, Raynaud phenomenon), previous corneal trauma or systemic disease affecting the cornea, and corneal surgery within 6 months of enrollment, were excluded. All participants provided written assent and parental informed consent, and the research adhered to the tenets of Declaration of Helsinki and was approved by the Weill Cornell Medicine-Qatar (WCM-Q) (19–00016) and Medical Research Center at HMC (MRC-01-20-761) Research Ethics Committees.

### Autism diagnostic criteria

ASD and ADHD were diagnosed by consultant pediatricians with a specialist interest in behavioral pediatrics, ADHD and ASD, according to the Diagnostic and Statistical Manual of Mental Disorders – version 5 (DSM-5) [30]. Individuals exhibited restricted, repetitive patterns of behavior, interests, or activities, such as stereotyped movements or speech, insistence on sameness, highly restricted interests, or hyper-or hyporeactivity to sensory input. Individuals with ASD met at least 3 of the criteria of social communication deficits and 2 out of 4 of the criteria for repetitive behaviour. The diagnosis of ADHD was based on DSM-5, which captured inattention, hyperactivity and impulsivity [31]. We did not have access to standardized diagnostic tools or questionnaires and data on diagnosis was extracted from the electronic medical records.

### Corneal confocal microscopy

Corneal confocal microscopy was undertaken using the Heidelberg Retina Tomograph III Rostock Cornea Module (Heidelberg Engineering, Heidelberg, Germany). One eye was anaesthetized with 2 drops of Bausch & Lomb Minims® (Oxybuprocaine hydrochloride 0.4% w/v). A drop of hypotears gel (Carbomer 0.2% eye gel) was placed on the tip of the objective lens and a sterile disposable TomoCap was placed over the lens, allowing optical coupling of the objective lens to the cornea. We acquired a minimum of 6 images per eye to allow selection of 3 images of the central sub basal nerve plexus (SBNP) with the best quality based on an established protocol [32]. Corneal nerve fiber density (CNFD) (fibers/mm$^2$) corneal nerve branch density (CNBD) (branches/mm$^2$), and corneal nerve fiber length (CNFL) (mm/mm$^2$) were quantified manually by HG using CCMetrics, who was blinded to the study group.

### Statistical analysis

As this was a pilot study, no formal sample size calculation was performed. Statistical analyses were performed using IBM SPSS Statistics software Version 30 and $P < 0.05$ was considered statistically significant. Normality of the data was assessed using the Shapiro-Wilk test and Q-Q plots. As data were normally distributed, they were expressed as mean ± standard deviation. Comparison between participants with ASD+ADHD, ASD and HC was performed using ANOVA with Bonferroni adjustment. Point estimates using Eta-squared ($\eta^2$) or Cohen's d and 95% Confidence Intervals (CI) were reported as appropriate. Adjusted analyses were performed using ANCOVA, controlling for age as a confounding variable. Graph prism version 10 was used to create figures.

### Results

Children with ASD had a DSM-5 score of 5.85 ± 1.26 and children with ASD+ADHD had a DSM-5 score of 5.92 ± 1.36. Age differed between ASD, ASD+ADHD and HC groups (15.0 ± 1.79 vs. 12.0 ± 3.21 vs. 14.1 ± 2.07, *P=0.02*) (Table 1).

**Table 1. Age, DSM-5 and corneal confocal microscopy metrics in patients with ASD, ASD+ADHD and HC.**

| Variables | HC (n = 15) | ASD-alone (n = 21) | ASD+ADHD (n = 6) | η² | 95% CI | P-value |
|---|---|---|---|---|---|---|
| Age (years) | 14.13 ± 2.07 | 15.00 ± 1.79 | 12.00 ± 3.21 | 0.18 | 0.003–0.36 | 0.02 |
| DSM-5 | – | 5.50 ± 0.84 | 5.92 ± 1.36 | 0.36 | −0.56–1.26 | 0.45 |
| CNFD (no./mm²) | 32.36 ± 7.17 | 27.08 ± 4.17 | 29.36 ± 7.45 | 0.07 | 0.0–0.22 | 0.24 |
| CNBD (no./mm²) | 73.47 ± 16.01 | 35.42 ± 16.46 | 49.50 ± 21.83 | 0.36 | 0.11–0.52 | <0.001* |
| CNFL (mm/mm²) | 23.37 ± 4.21 | 17.56 ± 3.66 | 20.02 ± 4.12 | 0.21 | 0.01–0.38 | 0.01* |

HC: healthy control, ASD: autism spectrum disorder, ASD+ADHD: autism spectrum disorder with attention deficit hyperactivity disorder; CI: confidence intervals; DSM-5: Diagnostic and Statistical Manual of Mental Disorders, Fifth Edition; CNFD: corneal nerve fiber density; CNBD: corneal nerve branch density; CNFL: corneal nerve fiber length. Bonferroni adjusment was used and P-value was significant if <0.016. Point estimate was reported as η²: Eta-squared for age, CNFD, CNBD and CNFL and Cohen's d for DSM-5.

CCM was not possible in three children with ASD+ADHD. CNFD did not differ between children with ASD+ADHD, ASD and HC (P = 0.24) (Table 1, Figs 1A, 2). CNBD was significantly lower in children with ASD+ADHD (P = 0.002) and ASD (P < 0.001) compared to HC (Figs 1B, 2). CNFL was significantly lower in children with ASD (P = 0.02) (Figs 1C, 2) with a borderline reduction in those with ASD+ADHD (P = 0.06), compared to HC. After adjusting for age as a covariate, the overall group effect on CNFD was not statistically significant (F = 2.18, P = 0.13), while both CNBD (F = 10.24, P=<0.001) and CNFL (F = 5.05, P < 0.01) remained significantly different between groups. Among the CCM parameters, CNBD and CNFL showed the largest group differences and were significantly lower in children with ASD and ASD+ADHD compared to HC, while CNFD did not differ significantly between groups. There was no difference in CNFD (mean difference = 2.28, 95% CI [−5.84, 10.40], P > 0.99) CNBD (mean difference = 14.1, 95% CI [−8.22, 36.4], p = 0.37) and CNFL (mean difference = 2.46, 95% CI [−2.28, 7.20], p = 0.61) between children with ASD+ADHD and ASD. DSM-5 did not correlate with CNFD (P = 0.876), CNBD (P = 0.405) or CNFL (P = 0.606).

## Discussion

This study shows evidence of corneal nerve loss in children with ASD+ADHD and ASD, extending our previous findings in children with ASD [22]. It confirms the presence of small nerve fibre damage, as evidenced by a lower intraepidermal

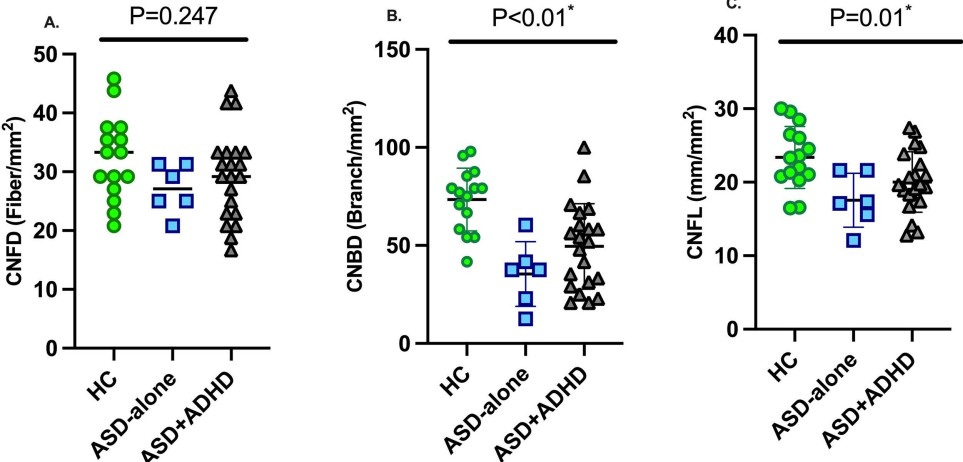

**Fig 1. CNFD (A), CNBD (B) and CNFL (C) in children with ASD+ADHD, ASD and HC.** HC: healthy control, ASD: autism spectrum disorder, ADHD: attention deficit hyperactivity disorder, CNFD: corneal nerve fiber density; CNBD: corneal nerve branch density; CNFL: corneal nerve fiber length.

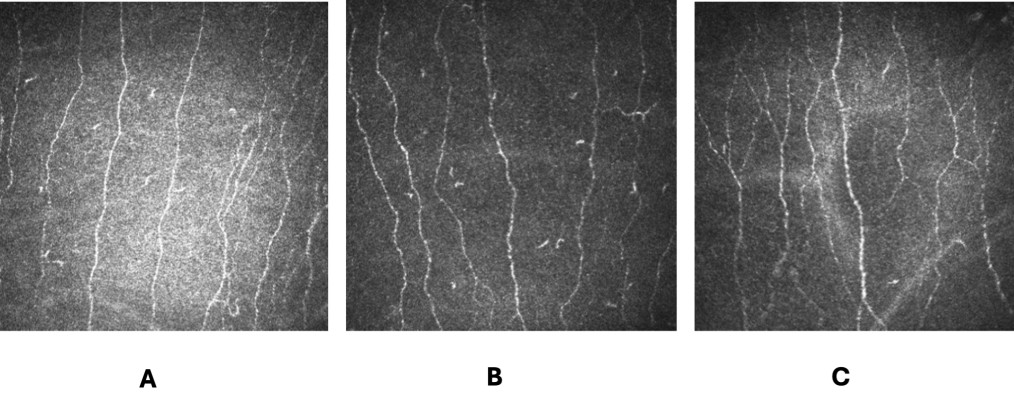

**Fig 2. CCM images in children with ASD+ADHD (A), ASD-alone (B) and healthy controls (C).**

nerve fiber density (IENFD) and elevated warm detection thresholds in adults with ASD [14]. These findings add to the evidence supporting the utility of CCM in identifying neurodegeneration in diabetic neuropathy [17], chemotherapy induced peripheral neuropathy, chronic inflammatory demyelinating neuropathy and HIV-neuropathy [18], dementia [19] and multiple sclerosis [33].

ASD and ADHD combined may exacerbate morbidity [34], but did not show evidence of greater neurodegeneration, although the cohort size was small, and we lacked a group with ADHD alone. Furthermore, direct subgroup comparisons between children with ASD+ADHD and ASD revealed no differences in corneal nerve fiber parameters, although the directionality of the findings indicated higher CNBD and CNFL values in the ASD+ADHD group. The lack of difference between children with ASD+ADHD and ASD could be attributed to the difference in age between groups as older age has been associated with a decline in corneal nerve function [35] and corneal nerve fiber density [36]. Additionally, treatment with stimulants such as Methylphenedate (MPH) may improve neuroplasticity and has been shown to modulate sensory thresholds in children with ADHD [37]. Moreover, both ASD and ADHD exhibit substantial heterogeneity in hyper- and hypo-responsivity, which may obscure differences [38].

There was no association between corneal nerve loss and the severity of ASD or ASD+ADHD, possibly reflecting limitations of the DSM-5-based severity scale, which primarily captures social-communication difficulties and restricted/repetitive behaviors, rather than underlying sensory deficits [39,40]. Atypical sensory processing is a non-specific marker of multiple neurodevelopmental conditions, and in the multicEntric Longitudinal study of childrEN with ASD (ELENA) [41] there were no signficant differences between children with ASD, ADHD and ASD+ADHD [23]. In a diffusion tensor imaging (DTI) study comparing adults with ASD and ADHD, voxel-wise fractional anisotropy (FA) and radial diffusivity (RD) were equally abnormal in the corpus callosum and related to sensory symptoms [42]. Participants with ADHD, self-reported greater tactile sensitivity and ADHD symptom severity, a lower tolerable threshold for electrical radial nerve stimulus and greater reduction in cortical SEP amplitudes to tactile stimuli which correlated with ADHD symptom severity [4]. A recent study in adults with ADHD demonstrated an increased risk of developing dementia [43], and we have previously shown corneal nerve loss in participants with mild cognitive impairment and dementia [19]. Neurofilament light chain (NfL), a marker of neuroaxonal damage is increased in a range of peripheral and central neurodegenerative diseases [44] including dementia [45] and is also increased in children with ASD [46], but not in adults with ADHD [47]. Furthermore, methylphenidate may be neuroprotective as treatment of children with ADHD showed enhanced spontaneous neural activity in the nucleus accumbens [48], whilst in animal studies it reduced mechanical and cold hyperalgesia and allodynia [49].

Limitations of this study include the small sample size of the ASD group and unequal group sizes, which may limit the statistical power. Differences in age between the study groups could also confound group comparisons. Specifically, the

effect size for CNFD was small, suggesting minimal difference, although CNBD and CNFL showed moderate effect sizes indicating detectable group differences. The lack of a group with ADHD limits our ability to determine whether corneal nerve loss is related to ASD or may also occur in ADHD. The lack of sensory testing and objective measures of ASD severity limits our ability to show an association with the severity of neurodegeneration. The lack of data on confounding factors such as concomitant medications, comorbidities, sleep disorders and ophthalmological disorders is also a limitation, as they may influence CCM outcomes.

These findings highlight the potential value of combining a comprehensive evaluation of motor coordination, visual–perceptual and sensory abilities alongside CCM into screening protocols to identify subtle neurodevelopmental difficulties in children with ASD or ASD and ADHD.

## Supporting information

**S1 File. Study raw data.**
(XLSX)

## Author contributions

**Conceptualization:** Adnan Khan, Rayaz A. Malik.

**Data curation:** Hoda Gad, Abdulla AlObaidi, Madeeha Kamal, Saba F. Elhag, Marian Aden, Adnan Khan, Areej Baraka, Srividya Bennabhaktula, Sara Joseph, Aisha Yaseen Alhamadi.

**Formal analysis:** Hoda Gad.

**Funding acquisition:** Adnan Khan, Rayaz A. Malik.

**Investigation:** Hoda Gad, Adnan Khan.

**Methodology:** Hoda Gad, Adnan Khan.

**Project administration:** Hoda Gad, Abdulla AlObaidi.

**Resources:** Rayaz A. Malik.

**Supervision:** Madeeha Kamal, Mohammed A. Tolefat, Abdulla A. Alhothi, Rayaz A. Malik.

**Writing – original draft:** Hoda Gad.

**Writing – review & editing:** Rayaz A. Malik.

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
