## [Decision Letter · Decision Letter 0]

9 Oct 2025

Dear Dr. Malik,

Thank you for submitting your manuscript to PLOS ONE. After careful consideration, we feel that it has merit but does not fully meet PLOS ONE’s publication criteria as it currently stands. Therefore, we invite you to submit a revised version of the manuscript that addresses the points raised during the review process.

We look forward to receiving your revised manuscript.

Kind regards,

Shivanand Kattimani

Academic Editor

PLOS ONE

2. In the online submission form, you indicated that the data underlying the results presented in the study are available upon request from the corresponding author.

4.Thank you for stating the following financial disclosure:

This study was funded by the Autism Research Institute

None.

6. Please amend your authorship list in your manuscript file to include author Mohammed A. Tolefat, Abdulla A. Alhothi.

7. Please amend the manuscript submission data (via Edit Submission) to include author Mohammed Tolefat, Abdulla Alhothi.

8. We note that there is identifying data in the Supporting Information file <19-000016_Initial Approval_03172020 (002).pdf>. Due to the inclusion of these potentially identifying data, we have removed this file from your file inventory. Prior to sharing human research participant data, authors should consult with an ethics committee to ensure data are shared in accordance with participant consent and all applicable local laws.

-Location data

Reviewers' comments:

Reviewer's Responses to Questions

**Comments to the Author**

1. Is the manuscript technically sound, and do the data support the conclusions?

Reviewer #1: Yes

Reviewer #2: No

Reviewer #3: No

2. Has the statistical analysis been performed appropriately and rigorously?

Reviewer #1: Yes

Reviewer #2: No

Reviewer #3: I Don't Know

3. Have the authors made all data underlying the findings in their manuscript fully available?

Reviewer #1: Yes

Reviewer #2: Yes

Reviewer #3: Yes

4. Is the manuscript presented in an intelligible fashion and written in standard English?

Reviewer #1: Yes

Reviewer #2: Yes

Reviewer #3: No

Reviewer #1: This is an interesting and potentially important pilot study investigating corneal confocal microscopy (CCM) measures of sensory nerve degeneration in children with ASD, with and without ADHD. The topic is novel, as few studies have examined peripheral neurodegeneration in pediatric neurodevelopmental conditions. The manuscript is generally well written and clearly structured, but there are several major and minor issues that should be addressed to strengthen the scientific rigor and clarity.

Major Comments

1. The study included 21 children with ASD+ADHD, 6 with ASD only, and 15 healthy controls. The very small sample size, especially for the ASD-only group, raises concerns about statistical power and generalizability. The authors should acknowledge this limitation more explicitly and discuss the implications for interpretation. A priori or post hoc power analysis would be helpful to clarify what effect sizes could reasonably be detected with the available sample, and to aid in the interpretation of null findings.

2. The design lacks a group of children with ADHD only, which limits the ability to disentangle whether the observed differences are specific to ASD or also present in ADHD. This gap should be acknowledged not only in the limitations but also in the rationale for study design, with an explanation of why the study remains valuable as an exploratory investigation. Where possible, the authors may cite existing literature on sensory nerve changes in ADHD.

3. Beyond age and sex, it is unclear whether other potential confounding factors were considered, such as medication use, comorbid conditions, sleep problems, or ophthalmological disorders. These factors could plausibly affect corneal confocal microscopy (CCM) outcomes and should be addressed either in the methods or the discussion.

4. Statistical analysis details

The manuscript requires more detail on the statistical procedures. Specifically:

(1)Were normality and homogeneity of variances checked?

(2)Given the small sample sizes, what statistical tests were applied (e.g., t-test, Mann–Whitney, ANOVA)?

(3)Was any correction for multiple comparisons (e.g., Bonferroni or FDR) performed?

(4)Were effect sizes or confidence intervals reported?

These details are essential for evaluating the reliability of the results.

5. Additional methodological detail on CCM protocols is warranted. For example, how many images were acquired per participant, and what criteria were used to select images (random vs. best quality)? Were images analyzed by a single rater or multiple raters, and was intra-rater or inter-rater reliability assessed? These factors are important for assessing the validity and reproducibility of the findings.

6. While statistically significant reductions in CNBD and CNFL were reported, the clinical meaning of these changes is not well elaborated. The authors should discuss whether the degree of reduction is likely to have functional consequences, and how CCM could potentially be translated into clinical practice for ASD/ADHD populations.

7. The lack of correlation between ASD severity scores and nerve measures is an important negative finding. This deserves more in-depth discussion. Does this suggest that sensory nerve degeneration is independent of behavioral symptom severity, or could it be related to limitations of the DSM-5-based severity scale?

8. The conclusion states that ADHD does not confer additional neurodegeneration in children with ASD. Given the small subgroup sizes (n=6 ASD-only, n=21 ASD+ADHD), such conclusions may be premature. The authors should temper their interpretation and frame the results as preliminary.

Minor Comments

1. Abstract: The abstract is clear but should briefly state the small sample size and the absence of an ADHD-only group to frame the findings appropriately.

2. Typographical correction in group labels: In the tables, the group name “ASD-AHDH” appears, which seems to be a typographical error for “ASD-ADHD.” Please correct this and ensure consistent labeling of groups across all tables, figures, and text.

3. Methods: The description of statistical tests should be expanded (e.g., correction for multiple comparisons? effect size reporting?).

4. Consistency of group labeling: The group names are not entirely consistent between the tables and figures (e.g., “ASD only” vs. “Autism,” “ASD+ADHD” vs. “Comorbid group”). The authors should standardize terminology across the text, tables, and figures to avoid confusion for readers. Using uniform labels (e.g., “ASD only,” “ASD+ADHD,” “Controls”) throughout the manuscript is strongly recommended.

In the tables, the group name “ASD-AHDH” appears, which seems to be a typographical error for “ASD-ADHD.” Please correct this and ensure consistent labeling of groups across all tables, figures, and text.

5. Discussion: The discussion could better integrate existing literature on peripheral nerve abnormalities in ASD (e.g., Chien et al., 2020, Neurology) and ADHD-related sensory processing studies.

6. References: Ensure all references are updated and formatted consistently.

Reviewer #2: Major Comments

Introduction

- Lines 76–78: The phrasing is awkward, as ASD and ADHD are “bundled together” in a way that blurs their distinct definitions. This opening sentence should be rewritten to introduce each condition clearly before addressing their overlap.

- Lines 80–82: The statement about atypical sensory features being a “core diagnostic criterion in DSM-5” is not entirely correct. While sensory peculiarities are explicitly acknowledged in the DSM-5 criteria, they were already well established in diagnostic tools such as the ADI-R and other scales. I agree that sensory issues are central in ASD, but the nuance is missed here: they are often underestimated (e.g., agitation misattributed to ADHD rather than sensory seeking) or overestimated (any sensory atypicality being interpreted as ASD). This complexity should be better reflected.

- Line 84: Rather than describing individuals as hyposensitive to subjective pain intensity, it would be more accurate to state that they show particularities in pain perception, as both hyper- and hyposensitivity are documented.

- Lines 96–97: The claim of corneal nerve loss in ASD relies on a single study authored by the same group. This should be presented more cautiously, with balanced language, rather than as established fact.

- Overall: The introduction is underdeveloped. It introduces non-consensual concepts (e.g., peripheral sensory nerve involvement) but does not defend them sufficiently. The innovative potential of these hypotheses needs to be articulated more clearly in relation to existing research that remains predominantly brain-centric. This contextualization would help readers understand why this line of inquiry matters.

- Lines 105–106: The manuscript lacks a clear hypothesis. Presenting corneal confocal microscopy (CCM) results in isolation risks being purely descriptive. The authors should articulate how detecting corneal alterations could improve understanding of patient experience, symptoms, or clinical care.

Methods

- The description of participants should clarify whether the ASD group included children with intellectual disability, as this is highly relevant to interpretation.

- Greater transparency is needed about recruitment strategy, representativeness, and diagnostic validation.

Results

- The small sample sizes (ASD without ADHD: n=6) limit statistical validity. Reporting a non-significant p=0.08 as evidence of no difference between groups is misleading. With such underpowered comparisons, absence of significance cannot be taken as equivalence.

General Assessment

The study proposes a potentially innovative shift from brain-centric to peripheral mechanisms in ASD/ADHD, but the introduction is too thin, the hypothesis is not clearly posed, and the interpretation is overstated given the sample size and reliance on prior self-citations. The paper would be strengthened by:

1. Rewriting the introduction for clarity and nuance (particularly around diagnostic criteria and sensory features).

2. Explicitly stating a testable hypothesis with clinical or mechanistic relevance.

3. Tempering claims where evidence is preliminary.

4. Expanding on the rationale for corneal measures—what do they add to patient understanding or care?

Reviewer #3: The authors are correct to highlight the fact that the changes to diagnostic categorisation of autism in the DSM-5 in 2013 resulted in a new separate criterion relating to atypical/unusual responses to sensory stimuli. Furthermore, prior to DSM-5, it was not possible to diagnose autism if ADHD were indicated. With advances in the literature, the common co-occurring nature of these two conditions suggests clinical research should address both presentations in single diagnostic groups and those with multiple diagnoses. This pilot study might contribute to knowledge in this field.

1. The authors need to be clear from the outset that this is a pilot study.

2. The abstract requires greater clarity about the aims of the study rather than simply what was done.

3. Terminology is unclear, i.e., when authors refer to neurodegeneration, do they mean loss of functioning/deterioration or ophthalmologic abnormality? The former would lead me to assume the pilot study aims to review participants over time rather than comparing clinical and non-clinical groups. Cf p2 lines 47 and 56 and p3 line 96.

Methodology

1. The current study description is not adequate to understand the processes or participant selection and hence to replicate the study.

2. Please capitalise the names of the recruiting centres and explain their remit/role.

3. Furthermore, how were the healthy controls selected?

4. How many families were approached before the final sample was selected? Did anyone drop out?

5. The authors have not explained participant characteristics, including whether the children had an existing diagnosis and, if so, how this was ratified, such as brief assessment, questionnaires or tasks. For instance, in lines 80-81, the authors state ‘there is an increasing recognition of abnormal sensory responses in ASD …which relate to quantitative autism traits… such that it now constitutes a core diagnostic [criterion] in DSM-5…’

6. There is no description of validation of the autism diagnoses, how children were identified or recruited, and the opportunity was lost to investigate any quantitative traits, such as those identified via completion of questionnaires and establishing whether the sensory differences criterion was met for all the autistic children and young people.

7. Symptom severity for both autism and ADHD require operational definitions.

8. The pilot study was small, with little detail about the range of participants, including cognitive ability.

9. Whilst age differences were not statistically significant, the non-autistic children were in general older than the autistic children. A table summarising the age ranges for all groups would be helpful.

Diagnostic criteria

1. It is not clear from this section whether the children were assessed and diagnosed as part of this study or whether they were recruited via a clinic where they already had received a diagnosis.

2. It is not noted how they were assessed, and which, if any tools supported the assessment and diagnostic process. Did the researchers have access to the clinical diagnostic assessment details?

Corneal Confocal Microscopy

1. Who conducted this?

2. At the end of the paragraph, it is noted that the investigator was blind to the images but does not say who conducted the procedure.

Discussion

1. P7 line 196 - This is not my field of expertise, but do the results demonstrate ‘loss’ or difference/pathology between groups?

2. P8 line 202 – refer to ELENA cohort earlier in the paper or explain it here.

3. The authors only identify two limitations, including the importance of objective measures of severity – this, in my mind is significant. I would argue that the findings are limited, and it is unclear what the next steps might be or implications of future research or indeed clinical impact.

Typos

1. P3 line 82 criteria should read criterion

2. P5, line 121, capitalise the full name of the manual and ensure that throughout the manuscript it is written correctly

3. P6 line 163 correlate

4. P7 Table 1 typos ADHD rather than AHDH

5. P8 ‘additonal’ should read additional?

**Do you want your identity to be public for this peer review?** For information about this choice, including consent withdrawal, please see our Privacy Policy

Reviewer #1: No

Reviewer #2: No

Reviewer #3: No

---

## [Author Response · Author response to Decision Letter 1]

22 Oct 2025

A point-by-point response has been uploaded.

---

## [Decision Letter · Decision Letter 1]

12 Nov 2025

Dear Dr. Malik,

Thank you for submitting your manuscript to PLOS ONE. After careful consideration, we feel that it has merit but does not fully meet PLOS ONE’s publication criteria as it currently stands. Therefore, we invite you to submit a revised version of the manuscript that addresses the points raised during the review process.

We look forward to receiving your revised manuscript.

Kind regards,

Shivanand Kattimani

Academic Editor

PLOS ONE

Journal Requirements:

Reviewers' comments:

**Comments to the Author**

Reviewer #1: All comments have been addressed

Reviewer #3: (No Response)

2. Is the manuscript technically sound, and do the data support the conclusions?

Reviewer #1: Yes

Reviewer #3: Partly

3. Has the statistical analysis been performed appropriately and rigorously?

Reviewer #1: Yes

Reviewer #3: I Don't Know

4. Have the authors made all data underlying the findings in their manuscript fully available?

Reviewer #1: Yes

Reviewer #3: Yes

5. Is the manuscript presented in an intelligible fashion and written in standard English?

Reviewer #1: Yes

Reviewer #3: Yes

Reviewer #1: The revision effectively clarifies methodology, improves the discussion’s conceptual integration, and adopts a balanced interpretation of results. Minor linguistic polishing and the addition of a brief statement on the clinical relevance of sensory and motor assessments in future screening would enhance the paper’s translational impact. Below are some minor suggestions.

1. Abstract and Introduction: The introduction is concise and sets up the rationale well. However, the research objectives would benefit from a brief mention that this is a preliminary exploratory study aiming to inform future mechanistic investigations and clinical screening strategies.

2. Methods: The methodological transparency has improved, but minor editorial polishing is needed for sentence flow. Clarifying how “routine clinical care” diagnoses were confirmed (e.g., by pediatric neurologist or psychiatrist) would strengthen credibility.

3. Results: the presentation of the results could be strengthened by providing greater quantitative detail and statistical context to assist readers in interpreting the effect magnitude and reliability. Specifically:

(1) Group comparison values (e.g., mean ± SD, p-values, or effect sizes such as Cohen’s d or η²) should be explicitly reported in-text or summarized in a table. This will allow readers to gauge the robustness of the observed differences beyond descriptive trends.

(2) It would be valuable to indicate whether the observed corneal nerve loss differed significantly between the ASD and ASD+ADHD subgroups, given the study’s stated aim of exploring comorbid ADHD effects. Even if the difference did not reach significance, reporting the directionality and confidence intervals would increase transparency.

(3) The authors may consider briefly stating which specific CCM parameters (e.g., corneal nerve fiber density, length, branch density) contributed most strongly to the group differences.

(4) Including a visual representation (figure or boxplot) summarizing group distributions could substantially enhance clarity and reader engagement, particularly for an exploratory study with small sample sizes.

4. Discussion: The discussion could be further strengthened by adding a short clinical implication statement, for example:

“In future screening and follow-up protocols, it may be valuable to include comprehensive evaluations of motor coordination and visual–perceptual abilities, allowing for the early identification of subtle neurodevelopmental difficulties that could precede or co-occur with psychosocial challenges in children with ASD or ASD+ADHD.”

5. Limitation and conclusion: The conclusion could end on a more forward-looking note, highlighting that future multimodal investigations combining corneal confocal microscopy with sensory and cognitive assessments could elucidate mechanisms of neurodevelopmental comorbidity.

Reviewer #3: 1. I continue to believe the authors have not clarified enough the likely benefit of the findings of the current pilot or potential benefits if a larger study were conducted. Despite the revision of the abstract I am still unclear.

2. lines 59 -60 ' Children with ASD+ADHD with and without ADHDASD-alone have evidence of sensory nerve

degeneration'. - typos

3. lines 86-88 'It is highly heterogeneous being varied in its presentation, co-occurring with conditions like intellectual

disability, and is shaped by factors such as age, gender, race/ethnicity, and developmental stage[1]' - this reads as thought ID commonly co-occurs with autism, whereas it is more likely that autistic people do not have additional ID. mental health conditions might be more relevant to add here.

4.Lines 95-98 'Most research has focused on brain centric mechanisms in both ASD and ADHD, however, there iss an increasing recognition of abnormal sensory responses in ASD [5-10] which relate to quantitative autism traits [7] such that it now constitutes one of the four sub-criteria under the restricted, repetitive behaviors domain a core diagnostic criteria in DSM-5 diagnostic criteria [11], but however this does no’t take into account the atypical sensory features of ASD' - this sentence should be rephrased and shortened/separated into two sentences and typos corrected.

5. 'Subjects with ASD' - check for consistency in terminology - subjects, patients or participants

6. ' Eligible participants with ASD were referred' - is this the case? did paediatricians actually refer patients or did they provide information about the research?

7. 'healthy controls were selected from the general pediatrics clinicstudy' -- this is unclear to me. Do all children regularly visit paediatric clinics for review or evaluation or are they referred if there are concerns? this may be different across international systems.

8. 'ASD was diagnosed as part of clinical care' - it still reads as though the children were assessed during the study - if they had already been assessed - i.e., some months or years earlier this should be clarified.

9. 'DSM5' to be written as DSM-5.

10. Overall, I am uncertain of the conclusions being drawn, and ability to replicate this study, given the methodology.

**Do you want your identity to be public for this peer review?** For information about this choice, including consent withdrawal, please see our Privacy Policy

Reviewer #1: No

Reviewer #3: No

---

## [Decision Letter · Decision Letter 2]

11 Dec 2025

Dear Dr. Malik,

Thank you for submitting your manuscript to PLOS ONE. After careful consideration, we feel that it has merit but does not fully meet PLOS ONE’s publication criteria as it currently stands. Therefore, we invite you to submit a revised version of the manuscript that addresses the points raised during the review process.

We look forward to receiving your revised manuscript.

Kind regards,

Shivanand Kattimani

Academic Editor

PLOS One

**Journal Requirements:**

Reviewers' comments:

Reviewer's Responses to Questions

**Comments to the Author**

Reviewer #1: All comments have been addressed

Reviewer #3: All comments have been addressed

2. Is the manuscript technically sound, and do the data support the conclusions?

Reviewer #1: Partly

Reviewer #3: Yes

3. Has the statistical analysis been performed appropriately and rigorously?

Reviewer #1: Yes

Reviewer #3: I Don't Know

4. Have the authors made all data underlying the findings in their manuscript fully available?

Reviewer #1: Yes

Reviewer #3: Yes

5. Is the manuscript presented in an intelligible fashion and written in standard English?

Reviewer #1: Yes

Reviewer #3: Yes

Reviewer #1: The revised version of the manuscript is much improved, with clearer presentation and a more focused interpretation. However, regarding the initial hypothesis that ASD+ADHD would show greater corneal nerve degeneration than ASD alone, the Discussion does not yet fully explain why this expectation was not supported. Beyond noting the small sample size and the absence of an ADHD-only comparison group, the authors could strengthen the Discussion by briefly considering several additional factors that may account for the non-significant findings:

1. Age differences between groups and developmental variation in corneal nerve parameters that could influence group comparisons;

2. The potential influence of ADHD neurobiology or stimulant medications, which may modulate peripheral nerve function and confound direct comparisons;

3. Heterogeneity in sensory-processing profiles across ASD and ADHD, which may result in distinct peripheral nerve patterns not captured by group-level analyses;

4. Interpretation of effect sizes and confidence intervals, to clarify whether the lack of significant differences reflects a true absence of group effects or insufficient statistical power.

The author may incorporate these points would provide a more comprehensive explanation for why the initial hypothesis was not confirmed and would help contextualize the findings within broader neurodevelopmental and sensory-processing frameworks.

Reviewer #3: I am satisfied that the authors have addressed the core concerns. I only have two minor typos to highlight:

p3 . It can vary in its presentation may be associated with anxiety and depression and shaped by age, gender,

race/ethnicity, and developmental stage [1]. COMMA MISSING

P4 Parkinson disease - Parkinson's disease

**Do you want your identity to be public for this peer review?** For information about this choice, including consent withdrawal, please see our Privacy Policy

Reviewer #1: No

Reviewer #3: No

---

## [Decision Letter · Decision Letter 3]

19 Jan 2026

Dear Dr. Malik,

Thank you for submitting your manuscript to PLOS ONE. After careful consideration, we feel that it has merit but does not fully meet PLOS ONE’s publication criteria as it currently stands. Therefore, we invite you to submit a revised version of the manuscript that addresses the points raised during the review process.

We look forward to receiving your revised manuscript.

Kind regards,

Shivanand Kattimani

Academic Editor

PLOS One

Journal Requirements:

**Additional Editor Comments:**

kindly address comments to authors by reviwer 3

Reviewers' comments:

Reviewer's Responses to Questions

**Comments to the Author**

Reviewer #1: All comments have been addressed

Reviewer #3: All comments have been addressed

2. Is the manuscript technically sound, and do the data support the conclusions?

Reviewer #1: Yes

Reviewer #3: Partly

3. Has the statistical analysis been performed appropriately and rigorously?

Reviewer #1: Yes

Reviewer #3: I Don't Know

4. Have the authors made all data underlying the findings in their manuscript fully available?

Reviewer #1: Yes

Reviewer #3: Yes

5. Is the manuscript presented in an intelligible fashion and written in standard English?

Reviewer #1: Yes

Reviewer #3: Yes

Reviewer #1: I am satisfied with the revised manuscript. Thank you for the authors’ efforts, and I look forward to seeing the paper published.

Reviewer #3: Thanks you for addressing the comments. I have two minor points to add with respect to your responses to Reviewer 1's comments.

Abstract needs to acknowledge the groups studied, particularly since an ADHD group alone was not included: 'a clinical screening strategy for neurodegneration in ASD and ADHD' should this read and ASD-ADHD combined?

Page 10 outlines the limitations of the exploratory study including group size. Since the groups differed with respect to age, should this also be added?

Many thanks

**Do you want your identity to be public for this peer review?** For information about this choice, including consent withdrawal, please see our Privacy Policy

Reviewer #1: No

Reviewer #3: No

---

## [Author Response · Author response to Decision Letter 4]

20 Jan 2026

Point-by-point response has been uploaded.

---

## [Editor Report · Decision Letter 4]

22 Jan 2026

Children with Autism Spectrum Disorder and Attention Deficit Hyperactivity Disorder have Evidence of Sensory Nerve Degeneration

PONE-D-25-44922R4

Dear Dr. Malik,

We’re pleased to inform you that your manuscript has been judged scientifically suitable for publication and will be formally accepted for publication once it meets all outstanding technical requirements.

Kind regards,

Shivanand Kattimani

Academic Editor

PLOS One

---

## [Editor Report · Acceptance letter]

PONE-D-25-44922R4

PLOS One

Dear Dr. Malik,

I'm pleased to inform you that your manuscript has been deemed suitable for publication in PLOS One. Congratulations! Your manuscript is now being handed over to our production team.

Kind regards,

on behalf of

Dr. Shivanand Kattimani

Academic Editor

PLOS One